# Caring for Caregivers: Italian Health Care Workers’ Needs during the COVID-19 Pandemic

**DOI:** 10.3390/ijerph182111386

**Published:** 2021-10-29

**Authors:** Diego De Leo, Maria Maddalena Martucci, Antonello Grossi, Francesca Siviero, Silvia Vicentini, Carolina Romascu, Arianna Mercurio, Martina Battaglia, Noemi Tribbia

**Affiliations:** 1Australian Institute for Suicide Research and Prevention, Griffith University, Brisbane, QLD 4122, Australia; 2Department of Mental Health, Local Health Unit 5, 45100 Rovigo, Italy; marilena.martucci@aulss5.veneto.it (M.M.M.); antonello.grossi@aulss5.veneto.it (A.G.); siviero.francesca@gmail.com (F.S.); dott.silviavicentini@gmail.com (S.V.); carolina.romascu96@gmail.com (C.R.); arianna.mercurio@yahoo.it (A.M.); martina.battaglia.3@studenti.unipd.it (M.B.)

**Keywords:** COVID-19, online survey, health workers, unmet needs, qualitative analysis

## Abstract

*Aim:* An online survey was proposed to the health workers of a public hospital of an Italian northern city. This was done with the aim of assessing the unmet needs of a special population under strain (the health carers) due to the fight against the coronavirus disease 19 (COVID-19). *Results:* By answering the survey, five hundred and nine people provided their observations, complaints and suggestions. This qualitative material was organised into three main areas: (1) relationship with the health organization management; (2) needs of the health workers; and (3) perceived consequences of the pandemic. *Discussion:* Overall, respondents expressed dissatisfaction for the unpreparedness of their health units and the confusion created by frequent changes in operational guidelines. Many participants felt abandoned, unheard and unprotected by the health organization, whilst the psychological support service formally set up by the hospital for its health workers was grossly under-utilised. Instead, support from colleagues and family constituted the main protective factor to counteract negative emotions. Restrictions in social contacts and recreational opportunities were frequently mentioned. Several respondents reported a sense of pride for their work and commitment; many others saw in the pandemic an opportunity for personal growth and better focus on important life values, like family and solidarity.

## 1. Introduction

On 11 March 2020, the World Health Organization (WHO) gave to the Sars-CoV-2 epidemic the status of pandemic; up to 11 July 2021, this caused the death of 4,027,861 people worldwide [1]. In Italy, from the beginning of the epidemic (first case: 20 February 2020) until 30 April 2021, the Integrated Surveillance System (ISS) recorded 4,035,367 positive coronavirus disease 19 (COVID-19) cases, with 1,867,940 happening just from January to April 2021 (46% of the total) [2] and 127.044 victims up to 11 July [3]. In Italy, the spread of the pandemic has followed a three-phase pattern. In the First Wave (March–May 2020), there was a huge spread of cases and deaths characterized by a strong concentration of contagions (mostly in the north of the country). In the Transition Phase (June–September 2020), there was a decrease in contagions and deaths. In the Second Wave (started at the end of September 2020), there was again a rapid increase in positive cases (until the first half of November 2020), then a decline (until December 2020), and a new increment commencing with the new year.

During these periods, with the aim of reducing the spread of the virus, various measures were applied, such as closing business and meeting places, imposing lockdowns in specific regions and local areas, etc. The outcome was a rift in everyday life activities and routines with an increased mental burden for the population [4] and a heterogeneous prevalence of the virus, even in nearby regions. For example, in Veneto (a region in northern Italy) the balance of deaths for all causes was contained in the First Wave (+19.4%), while the close Lombardy recorded the highest rates of the country (+111.8%) [5]. However, during the first part of the Second Wave, Veneto recorded an increase in mortality as well (+44.4%) [5]. This was attributed to the decentralized organization of the Italian health-care system and the adoption of different local policies of containment [6,7].

### 1.1. COVID-19 and Mental Distress of Health-Care Workers

In this scenario, health-care workers (HCWs) had to cope with various types of risk factors for physical and psychological distress, due to the exposure to the contagion and the unprecedented pressure of hospital services. Right from the beginning, the HCWs experienced a stressful increase in workload, linked to the procedures of management and notification of COVID-19 cases and organizational consequences—such as enlargement of intensive care units, redistribution of tasks beyond professional specialization and changes in access to visits, admission criteria, daily practices, etc. (see, for example, [6,8] for descriptions of Italian scenarios). Furthermore, HCWs reported problems about personal protective equipment (PPE), like shortage and forced reuse of single-use PPE items, low confidence in their use due to lack of training from the organization [6,8,9,10,11,12]; adverse physical effects at work (headaches, exhaustion, thirst, etc.), perception of risk of infection and fear of spreading the virus, mostly in the family [9,11,13,14,15,16,17,18]; and, distress caused by the improved knowledge of the dangerousness of the virus and its possible consequences [19]. In the Italian context, Puci et al. [18] showed that most of HCWs perceived a very high risk of infection (about 60% at the beginning and in the end of the First Wave) together with a greater risk of it in the workplace than outside (70.98% vs. 5.68% at the beginning and 48.18% vs. 17.95% in the end of the First Wave). Savoia et al. [11] highlighted that most physicians perceived a significantly higher risk of being infected at work (rating a risk of 56 out of 100) than in other contexts (risk of 25 out of 100) and they were adopting precautions to prevent their family infection (72%). Then, in the end of the First Wave, Rapisarda et al. [14] highlighted that the HCWs in contact with patients reported an average probability of infection at work of 59.8% and were quite worried of being infected (51.7%). Then, Costantini et al. [8] showed that even hospice organizations were worried about the risk of their staff becoming ill by COVID-19, reporting greater average values in medium (6.2/10) and high prevalence areas (4/10) than in low prevalence ones (2.5/10).

In addition, continuing changes in the hospital protocols due to the COVID-19-pandemic evolution significantly contributed to the increased risk of psychological distress in HCWs [20]. During pandemic time, HCWs’ mental health was subject to developing phenomena like anxiety, depression, insomnia, high levels of stress and post-traumatic stress disorder (especially with work performed in high-risk areas) [21,22,23,24]. Cross-sectional studies on COVID-19 pandemic around the world [15,25,26], and Italy [14,19,27,28] have confirmed these observations.

The role of job position is quite unclear. Some studies highlighted that nurses would experience more mental health symptoms than doctors [21,22,23,24], but Zhang et al. [26], for example, found the opposite. In Italy, Rapisarda et al. [14] reported that doctors and HCWs of outpatient services were respectively more vulnerable to burnout than nurses and HCWs of inpatient services, while Simione and Gnagnarella [19] found that HCWs from northern regions felt more stress and anxiety than their colleagues in the rest of Italy (possibly in relation to the higher prevalence of registered cases of COVID-19).

### 1.2. Health-Care Workers and Pandemic: Protective Factors

Despite the negative outcomes of the pandemic, researchers found also that HCWs could identify protective factors able to reduce the level of stress or help them to cope with the pandemic, such as: perceived social, professional and moral responsibility; expectation to receive adequate recognition from the government and the health organization; availability of clear institutional protocols and guidelines to avoid infection and/or its management; self-care coping and management styles (including psychological adjustment, keeping informed about COVID-19 prevention and transmission, isolating themselves, taking care of the working team, etc.) and various forms of social support (e.g., from colleagues, family, friends and society) [9,13,17]. Specifically, Cai et al. [13] and Liu et al. [9] highlighted that support from family and friends and perceived optimism in the work environment help HCWs to feel less lonely and more motivated during their work.

Good management by health organizations would represent an important protective factor for HCWs. However, especially the initial phases of the pandemic have shown a substantial unpreparedness of the health-care services both in Italy (e.g., [8]) and elsewhere [29]. Lack of preparedness has been confirmed by a qualitative study of Nyashanu et al. [10] on the personal narratives of domiciliary frontliners that complained about delays in performing tests, general absence of clear policies to deal with COVID-19, and the chaotic guidance of public health authorities and government.

Considering this critical scenario, in the context of a project for the prevention of depression and the promotion of public health, the Department of Mental Health of the Azienda ULSS-5 Polesana (AUP) in Veneto Region decided to investigate with a survey the impact of the COVID-19 pandemic on the HCWs employed in the health-care services of the province of Rovigo (a Veneto city, north of Italy), asking everyone to express their personal and professional needs during the pandemic and how they lived this period. The overall aim of the survey was to explore the main difficulties, negative emotions and coping strategies of the HCWs employed in the AUP, and their reflections and conceptions about the pandemic status.

## 2. Materials and Methods

### 2.1. Study Setting and Sample

Within the project “The prevention of suicidal phenomena—a General Imperative” (supported by the bank *Fondazione Cassa di Risparmio di Padova e Rovigo*), a questionnaire entitled “The impact of the pandemic in the life stories of health-care workers” was administered online. The questionnaire was disseminated via corporate emails, wall posters with a QR code and posted on the bulletin boards of common areas of the health-care services. All the questionnaires were to be completed anonymously, and the data were collected between 22 December 2020, and 21 March 2021. All participants consented to take part to the study without reward or constriction, and freely gave their informed consent to data collection and analysis. During the time of the investigation, we had the approval of the Director General of the Local Health Unit which implied that all the requirements to perform the research had been fulfilled (approval number: prot. no. 0116438/VII.5). The questionnaire was designed inside an online survey, which had a main quantitative part (with 18 close-ended questions), and a secondary facultative qualitative part (with two open-ended questions). The adopted questionnaire collected main demographic data and the professional and psychological experience of HCWs in Italy during the pandemic. The qualitative part had the aim to better contextualize the results of the quantitative part. For this purpose, in the qualitative part it was explored: (a) the personal and professional needs of the HCWs during the COVID-19 emergency, and (b) their personal and professional reflections and conceptions about the pandemic. To avoid burdening the survey, the open-ended questions were formulated in a general way (indeed, various specific areas were investigated through the close-ended questions: e.g., the most perceived negative emotions or the most used coping strategies). Among those who accepted to join the survey, we considered as eligible for the analysis only those AUP’s HCWs who answered to the two open-ended questions, which were the focus of this paper.

### 2.2. Data Analysis

The responses obtained by the open-ended questions were analyzed through the software Atlas.ti [30]. The software allows users to identify the fundamental themes emerging from the narratives (through repeated words or synonyms, or through similar phrases or concepts) and to cluster them in specific codes and macro-codes, in order to capture the sovra-ordinary meanings expressed in the writings. In light of this, a thematic analysis was made on every open-ended answer adopting a bottom-up approach derived from the Grounded Theory, a perspective that invites the researcher to be reflective in the interaction with the data and infer the theory directly from them [31,32]. In this theorical framework the thematic analysis is the instrument with which the researcher can generate analytical categories and identify their reciprocal relationships. The most significant quotes were reported in-text specifying the number that identifies the participant (from 1 to 509) and the number that identifies the found code. In this article, names are fictitious in order to prevent any possible identification of the participants.

## 3. Results

Among the 3084 HCWs employed in the AUP, 749 individuals accepted joining the study. The questionnaire presented with both closed and open answer opportunities, and among them 509 participants (mean age = 48.07 years; SD = 10.97) answered the open-ended questions. Their characteristics are reported in Table 1. They were described in a clustering of the demographic data through the following categories: sex, ten-year age groups (from 20–29 years to >69 years), working area in the AUP organization, educational level, having children, and social status. At the time of the online posting of the questionnaire, only 9 health workers (1.77%) completed the questionnaire before the first dose of COVID-19 vaccine, 500 (98.23%) after it.

The qualitative analysis of the questionnaires revealed several codes that were divided into families according to the expressed topic. Specifically, three main topics were identified concerning: (1) the relationship with the AUP during the pandemic, (2) the needs of HCWs; and (3) perceived consequences of the pandemic. Each topic was then subdivided into subtopics based on the issues of greater relevance.


**Theme 1: Relationship with the AUP**


**1.1** 
**AUP Unpreparedness**


Many participants pointed out that, during the health emergency, the health institution was unprepared to deal with it and this led HCWs to feel the need for a more organized level of management and notification of COVID-19 cases. Like Stefano explained, *“One thing that strikes me is the demonstrated inability to manage emergency situations even by people/authorities paid or appointed precisely for these eventualities. In essence, the inconsistency, fragility and managerial inadequacy (at all levels) of the command hierarchy is striking”* [236:3]. The lack of clear protocols to avoid the danger of being exposed to infection or to expose others, the lack of all the PPE needed, and the absence of any formal training to learn how to deal with patients, were particularly remarked. As Giovanni said, *“There should be a clear planning of the actions necessary to protect the safety of those who try to help those who got ill (e.g., a specific triage, etc.)”* [161:3].

**1.2** 
**Lack of Organizational Support**


Many HCWs have experienced a lack of opportunity to have constructive discussions with their managers and had difficulties in collaborating with their colleagues, highlighting a shortage in the number of staff and, therefore, a heavy workload. Like Amelia pointed out, *“The pandemic has certainly highlighted the weaknesses of political choices made in the field of health, regarding undue financial cuts and unclear policies at the community level. We need to invest more in health care, decentralize activities, hire more staff, increase the salaries of employees that have remained at the same level for too many years”* [200:4].

As shown in Figure 1, all the highlighted factors led to perceived psychological and managerial abandonment by the AUP. Like Chiara stated, *“The distress I feel during this situation is mostly at the organizational level: where I’m working I don’t feel protected or supported. The organizational system is poor and one feels alone in facing such an emergency”* [306:2]. This situation elicits feelings like frustration, anger, stress, and anxiety. For example, Eleonora stated: “*I’m disappointed and angry with the system: I feel used, without any recognition”* [437:1]. The final evidence that emerged within this theme is the positive consideration with which this questionnaire was received: some participants have underlined their appreciation for this initiative because they felt considered and listened to by the institution, like Rebecca: “*Thank you for this opportunity, having the possibility to talk about own experiences is essential”* [48:1].

**1.3** 
**Positive Sides of the Work Environment**


Otherwise, some participants expressed the pride derived from doing the job and being able to help ill people, despite the difficulties encountered. Arianna said, “*I am happy to have had my part in this war: for three months I have worked in the Intensive Care Unit and every day I live two conflicting feelings inside me. On the one hand, death: defeat, rage, loss of a patient without the presence of those who loved them (I represent them!), frustration, sadness and crying, because I have assisted them in the previous days; on the other hand, tears of joy when a patient is discharged, we won against the virus, what a joy! In those moments my heart is bursting!”* [137:6].

Some participants talked about the support they perceived from colleagues during the hard times, like Andrea: “Of course, if you have the fortune to work surrounded by wonderful colleagues like mine, there is already a reason to be grateful despite the horrible period” [495:2].


**Theme 2: Needs of the HCWs**


**2.1** 
**Psychological, Concrete and Professional Needs**


Considering what emerged from the first thematic, many HCWs have expressed the need to have some form of salary recognition for the job carried out and the necessity of more serenity and optimism inside the work environment; like Scarlet said: *“I would need support with other resources and more recognition, including financial, from the institution and our managers. [...] I would also like to feel less tension in the workplace”* [479:1]. Moreover, being able to rest and have some vacations was reported by a few people, such as Viola, who said: *“With the second wave, all recreational leaves were suspended, without considering that the pandemic was not only here, at work, but it was everywhere, in our homes too, and we needed those damn vacations also to manage the family and give a rest to our minds from those battlefields with few survivors”* [308:3].

Several HCWs also expressed the fear of testing positive to COVID-19 or infecting their family members, leading them to express the need for both greater public awareness of the regulations implemented against the virus and greater scientific knowledge of how the virus is transmitted. For example, Aurora observed: *“This period is lived with the stress and the frustration from the daily routine, the fear of infecting others, especially our families as well as our colleagues, despite the attention and concentration always present in everything we do. All this becomes stress and physical and psychological fatigue that render difficult to perform our work and daily activities with tranquility, even if we try to give the best of ourselves in our service”* [135:3].

Some participants, like Penelope, also expressed the need to receive professional help to cope with the experienced difficulties: *“I would also need a professional with whom I could vent my feelings instead of just working and continually repressing fear and pain.”* [173:2].

A group of people specified that they had no need or did not know what that need could be, like Ginevra, who said: *“I wouldn’t know what would help in this difficult situation”* [218:2].

**2.2** 
**Perceived deficits**


Many people expressed that they miss their loved ones, physical contact, and even the ability to pursue their hobbies given the restrictions imposed by the government. Edoardo stated how these conditions led to a perceived lack of being able to share concerns and tension of daily life without receiving support from anyone: “*I would need to resume a normal frequency of social interactions, being off-site and going elsewhere, to another region. I have suffered greatly from not being able to see my loved ones for months. And this has been going on for too long, now”* [511:1].

For a better understanding of the links found between the various codes in this theme, Figure 2 was created.


**Theme 3: Perceived Consequences of the Pandemic**


This topic concerns thoughts and reflections that HCWs expressed about the pandemic.

**3.1** 
**Uncertainty: Source of Distress or Chance for Growth**


Considering uncertainties regarding the future that characterizes this historical period, many participants expressed the need to know when the pandemic would be over in order to return to normalcy; like Francesco said: “*Overall, I was less of a victim of isolation than many. The position of health care provider allowed me a certain amount of autonomy. It also allowed me to get involved in providing professional and psychological support to patients, family and friends. Personally, I suffer from travel restrictions and the fact that this condition has been going on for a year, as well as the uncertainty of what will happen tomorrow. I have great confidence in a vaccination campaign that can bring the situation under control and ensure a return to normalcy in an acceptable timeframe*” [511:2]. Then, many HCWs made comments about the way the government handled this emergency situation and the way the media had communicated the information about COVID-19. Paola stated: “*I believe that a lot of confusion has been created; that there has been a lack of organization and logic in legislative choices at all levels; that the media has exaggeratedly frightened people and that the right to freedom and free will has been threatened without giving clear or reassuring messages, making only appeal to people’s fears. I believe there has been no much common sense and much confusion*” [241:1].

However, some HCWs seemed capable to see in the pandemic a chance also for some positive growth. Matteo said: “*This is a new experience for everyone. Not even epidemiologists have ever witnessed such a global event. So, doubts, mistakes, and in itinere corrections are normal. At the same time, I recognize the great opportunity to evolve, change and improve. I am disheartened by the prevailing individualism, the intolerance often linked to comfortable previous conditions and the ease in delegating responsibility, but it is the physiological course of unique events. I only hope that some of us can come out of it as better persons*” [19:4].

In this sense, as depicted in Figure 3, some participants reported positive strategies they implemented in order to cope with the pandemic, which led them to experience a state of well-being and to express hope for positive growth as an outcome of the situation. For example, Ilaria said: *“In this period I have understood many things: the beauty of life, the desire to do things and change; I have rediscovered people in their fragility and inability to accept restrictions”* [353:2]. Irene said: *“Today I think I have gotten used to this new* “modus operandi*”, this way of living and working that is so different from what it used to be. Personally, I’m fine, and I’m strengthened by the thought that in the pandemic misfortune I was lucky: I did not get sick and neither did my family. We are all doing well, and this increases a sense of gratitude, courage and strength to keep going with passion and love”* [63:3].

**3.2** 
**Consequences for Ill People and Their Families**


Lucia reflected on the condition that hospitalized patients and their families experienced: “*The pandemic at first made us vulnerable, but at the same time it strengthened us, in some cases even improved us as people. The realization that a patient is isolated, deprived of all his family contacts with the risk of never seeing them again and* vice versa *has made me reflect a lot. This is something I can’t digest: losing a family member without being able to comfort them, stay close to them and not give them a final goodbye is simply unacceptable*” [458:2]. In addition, some have brought to attention the need to restore the original organization of care for all other diseases, suspended by the health emergency, both at the community level and hospital. Nicole said: “*It is necessary to ensure the possibility of treatment and prevention even in the presence of COVID-19. Now you cannot do prevention, specialist visits, therapies. It is pure madness that everything is focused only and exclusively on COVID-19. Those who have other pathologies feel abandoned and are worried about their future*” [141:4].

## 4. Discussion

This study proposed a thematic organization of the personal and professional needs and experiences reported and described by the HCWs of the province of Rovigo (northern Italy) during the Second Wave of the COVID-19 pandemic.

Regarding their COVID-19 psychological and professional experience, the results of the survey’s quantitative part showed that many HCWs reported feelings of uncertainty (40.16%), fear that the people they care about may get sick (65.47%) and worries of being infected (27.03%) as the main pandemic difficulties. A global sense of abandonment and personal unpreparedness emerged. Most HCWs declared they adopted useful coping strategies to deal with pandemic difficulties (such as, to do the things they like most under the restrictions: 53.44%) and seemed able to receive support from their social network (e.g from family—34.84% and colleagues—25%). Some of them declared that with the pandemic they rediscovered some neglected values—such as the importance of the relationship with family and friends (30.47%) or the value of life (36.72%). HCWs globally reported negative emotions (such as frustration—32.66% or helplessness—40.94%) and increased distress due to the heavy workload (especially frontliners: 45.74%) and the changes in life-style due to the self-isolation (27.81%). In their answers to the open-ended questions, they explored deeply these themes highlighting: (a) why they perceived organizational abandonment; (b) the positive aspects of working during the pandemic; (c) the needs and deficits they perceived during periods of distress; (d) how they cope with the feelings of uncertainty; (e) which personal and social pandemic consequences they perceived.

Specifically, a first theme named “Relationship with the AUP” was identified. This theme encompasses the emotions felt by the HCWs, the deficiencies and unpreparedness perceived as deriving from the health organization, but also the perceived support and the pride derived from the work showing various similarities with the pandemic experiences of other HCWs from Italy and other countries. Specifically, we found numerous references to the substantial unpreparedness of the health organization and its impact on HCWs’ work conditions, satisfaction and wellbeing. Many participants expressed dissatisfaction for the confusion created by continuous changes in guidelines. This is in line with the study of Costantini et al. [8], who identified a general lack of training and clarity by the institutional managers of Italian hospitals. Several researches from Europe and Asia claimed that physical and psychological distress in HCWs was mainly caused by a perception of disinterest by the managers, not providing concrete (e.g., availability of PPE) and emotional support [8,10,17]. According to these findings, most participants felt abandoned, unheard and unprotected by their health organization; these aspects led them to feel anxious, depressed, distressed, disappointed and at risk of mental health problems [25]. Notwithstanding the general frustration, HCWs were able to implement own coping strategies to deal with the pandemic thanks to the colleagues’ support (which was the main protective factor to counteract negative emotions and mental health problems, together with family support) [9,13,17] and professional pride. Indeed, according to Sun et al. [17], many references about feelings of pride for one’s own work and profession were found in the writings, suggesting professional pride as a common experience during the COVID-19 pandemic among HCWs worldwide. In the responses, these feelings generally co-occur with optimism and gratitude to having been on the front line against the pandemic.

The second identified theme, “Needs of the HCWs”, encompasses all psychological, concrete and professional needs the HCWs expressed about their stressful work conditions and fears. Specifically, many people expressed the desire to work with less pressure and complained about the lack of rest or vacation, causing less concentration during the job. This evidence is coherent with previous literature, pinpointing that excessive workloads were identified as the main risk factor for burnout and physical and mental exhaustion of HCWs [33], as also detected during the COVID-19 pandemic elsewhere [22,27]. Moreover, the fear of being infected or to infect family members caused many difficulties in maintaining interpersonal relationships. Participants expressed the need to visit their partners, relatives and friends or to be reunited with their families when they lived in another city or region. As seen in Tomaino et al. [4], disruption in social life was pointed out by many Italian people, noting suffering from the absence of physical contact with their loved ones. This fact, along with the other imposed restrictions, led the people interviewed to feel limited in social life, in cultivating their hobbies, and in making plans freely [4]. This could explain why the need of freedom was frequently expressed by the HCWs in the present study.

Despite the evidences reported above, only a few people expressed the need for professional support. This might seem counterintuitive, considering the high level of psychological and physical distress reported. In trying to explain this apparently contradictory finding, it could be hypothesized that the scant exigence of professional support could be the consequence of feared stigmatization about consulting a psychologist or, simply, that the support of loved ones was preferred to professional help. As a matter of fact, at the beginning of the pandemic the health institution created a psychological support service for its health workers; however, that service ended by being poorly utilized (during the time of the investigation, only six persons requested professional support from the service).

Some participants did not express any need. This finding did not appear related to any specific demographic characteristics (such as age, sex, marital status or workplace); so, it was hypothesized that these people had no serious difficulties in adapting to the pandemic situation or did not suffer from the social isolation. These interpretations are only conjectural and would need to be confirmed with deeper interviews.

Finally, the third topic (“Perceived consequences of the pandemic”) was helpful in contextualizing the needs and the deficits previously exposed. The main evidence that emerged concerns the pandemic as an opportunity for change. Some respondents reported greater awareness for what is really important in life, showing an increase in self-reflection, which is an experience highlighted even by Sun et al. [17] on Chinese HCWs and Tomaino et al. [4] on the general Italian population. This could suggest that an increment in self-reflection is a common experience during the pandemic for everyone. Specifically, the HCWs stated that the pandemic did not have only negative outcomes, but also gave people the chance for positive growth [4], with many of them recognizing the importance of conditions such as good health and having a family [4,17]. Finally, some respondents expressed anger for the level of isolation of ill people and the distress they have lived, or for the disinformation and fear that the press caused. Bad information, contradictory news or even fake news had a real responsibility in creating and feeding fears and worries during epidemics [16]. Someone also highlighted the perception of abandonment in people suffering from other pathologies and many of them, in line with Cipolletta and Ortu [34] and Tomaino et al. [4], reported painful feelings of uncertainty for the future.

## 5. Conclusions

As was pointed out, the HCWs delt with the pandemic in different ways. Many respondents lived throughout that period not accepting the restrictions and the changes in work and life imposed to all by the pandemic. Most respondents experienced the pandemic in a traumatic way, where routines and previous life habits were destroyed. Some expressed concern for people affected by other illnesses who were left behind because of pandemic priorities.

Regardless, in some participant’s narratives, it is possible to recognize reference to the pandemic as a chance for positive growth (such as the rediscovery of the many fragilities in life and the importance of solidarity) and increment in self-reflection—common experiences during the COVID-19 crisis both in general [4] and HCWs population [17]—but even feelings of pride in one’s own work and profession, as HCWs of other countries had reported [17].

These views are complementary. In literature, they are reported as common outcomes of the pandemic [4,34]. Many HCWs expressed gratitude for the opportunity of being interviewed through a questionnaire: they felt listened by someone from the health organization. Being unheard is associated with burnout, a negative outcome that affected HCWs worldwide during the pandemic [27].

This study has several limitations. The most relevant one is represented by the relatively small number of respondents (out of the 749 individuals that answered the complete questionnaire, only 509 provided qualitative answers. The total number of employees of the health organization, including administrative personnel and cleaners, was 3084). Then, it was an online survey, and the participants did not have the possibility to ask for clarification for eventual doubts about the questions. This might have led to inaccurate or too vague responses. The themes identified in respondents’ narratives are quite vast and inevitably include heterogeneous statements. For these reasons, this study should be considered as a starting point for future researches. Despite its limitations, our investigation might have value in representing the first example of qualitative observations on a rather large sample of health workers of an Italian public setting during the COVID-19 crisis.

## Figures and Tables

**Figure 1 ijerph-18-11386-f001:**
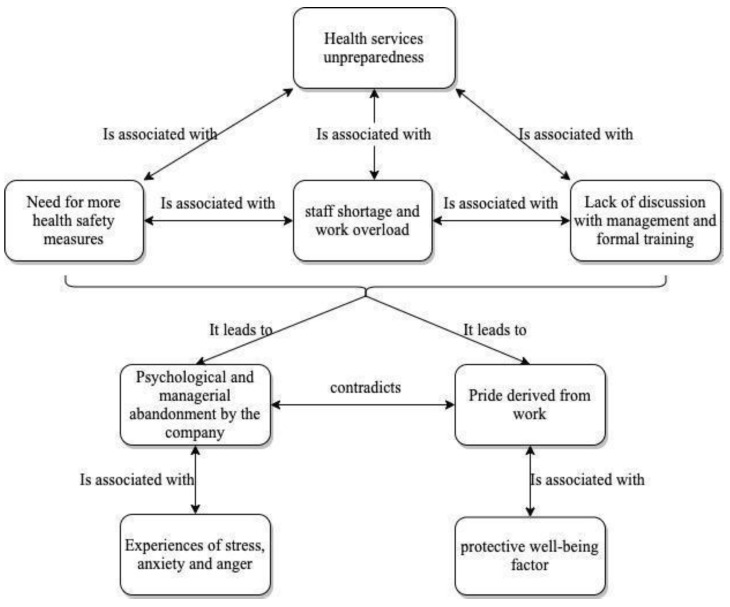
Relationship with the company.

**Figure 2 ijerph-18-11386-f002:**
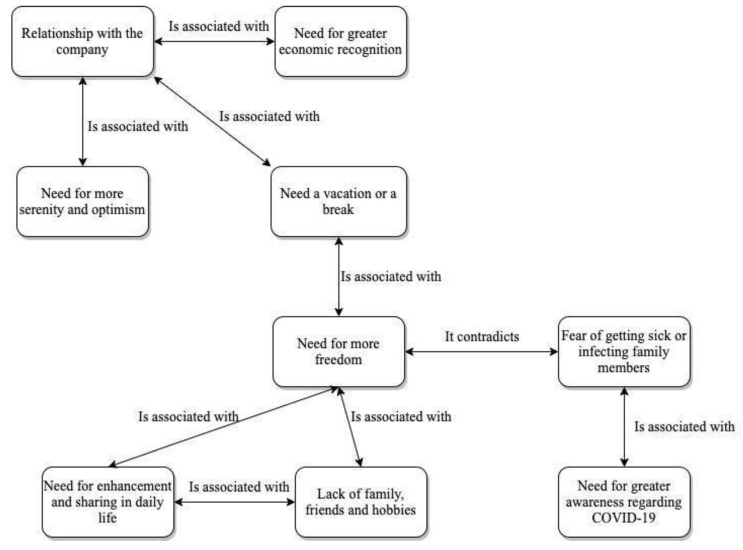
Needs of the HCWs.

**Figure 3 ijerph-18-11386-f003:**
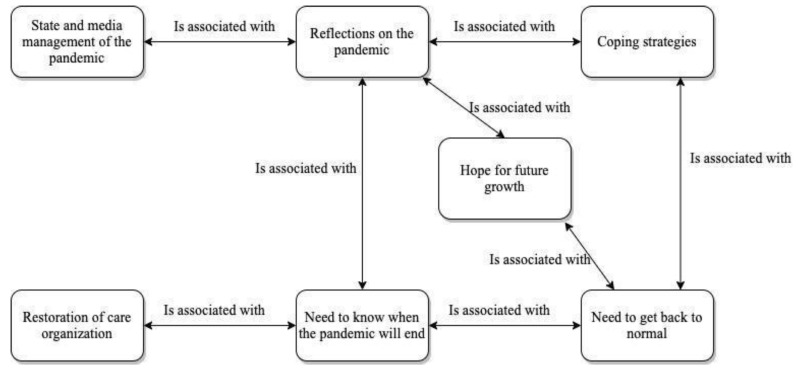
Perceived consequences of the pandemic.

**Table 1 ijerph-18-11386-t001:** Participant characteristics (*n* = 509).

Characteristics	No.	%
Date of completion		
Post-vaccine	500	98.23
Pre-vaccine	9	1.77
Sex		
Men	130	25.54
Woman	379	74.46
Age		
19–29	36	7.07
30–39	68	13.36
40–49	112	22.00
50–59	224	44.01
60–69	66	12.97
>69	1	0.20
Working position		
General ward	170	33.40
Territorial services	178	34.97
COVID-19 ward	81	15.91
Administrative services	73	14.37
Not working in AUP	6	1.18
Educational level		
PhD	7	1.38
Post-lauream master	68	13.36
University	242	47.54
High School	170	33.40
Middle School	21	4.13
Children		
Yes	330	64.83
No	178	34.97
Social status		
Married	293	57.56
Involved with someone	84	16.50
Single	60	11.79
Divorced	60	11.79
Widowed	11	2.16

## Data Availability

The data presented in this study are available on request from the corresponding author. The data are not publicly available due to data sensibility.

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
