# Peer review of "Caring for Caregivers: Italian Health Care Workers’ Needs during the COVID-19 Pandemic"

_ijerph, 2021, doi:10.3390/ijerph182111386_

Round 1

Reviewer 1 Report

Dear authors,

the research you have presented is interesting, however changes need to be made in order to improve its methodological robustness. Therefore, please refer to the suggestions below.

1.Introduction

Line 63-65: About risk perception of being infected, please consider improving the introduction with data from the Italian context. In the literature, several surveys have studied risk perception among Italian HCWs.

Study aim

Please note that the study aim is the backbone of the paper, as it is essential for defining study design and data analysis. The results should be aimed at its achievement, which will then be discussed in the discussion and conclusion paragraph. Therefore, the aim should be clear and reachable. For this reason please consider revising the aim by changing “to give voice “into something that is more defined and scientific soundness (such as the “to describe, to explore” etc. ...).

2.Methods

This paragraph has some important incorrect aspects from a methodological point of view. However, they should be revised and improved. Therefore, please consider the following advices.

  • Please add information about study design/research.
  • Mean age of the sample is a result and it should be moved from methods section to the results one. Authors should be the same for other results (number of participants, % of HCWs filled out the questionnaire after or before vaccines). Table 1 shows characteristics of the sample, these are descriptive results (add also techniques used to summarize them); please move this table in the results section. In addition, the denominator in the table rows is redundant as this data is already present in the first row remove it (example: sex, men 130/509), please also add mean age (sd) and the total number in the title of the table.
  • Did the study receive the approval of an ethics committee? If the answer is no, this aspect should be justified.
  • Please add eligibility criteria (i.e.) and if it is possible justify information about Sample size.

Reviewer 2 Report

Even though a qualitative approach to the topic seems appropriate, the ways in which te study was conducted and is presented do not meet such expectations.

First, the reader cannot get a clear idea of the main goal of the study (impact, needs, experiences? of HCW). This gets worse when the open-ended question whose answers are being analysed is not even presented in the manuscript.

A second major flaw is that the study did not go through any Ethical Committee or, at least, the manuscript does not report this.

The data analysis is presented a bit confusingly: was it a thematic analysis or a grounded theory study? In any case, the results are presented under too broad categories (e.g. reflections on the pandemic), which does not help getting a clear sense of the focus of the study.

Finally, the discussion and conclusions refer to a sort of "narrative" and phenomenological (i.e. research on the lived experience of the HCW) approach which is not consistent with the methods and results previously described.

Even though I am not a native English speaker myself, I reckon a thorough revision of the language used in the manuscript is needed.  

Reviewer 3 Report

This is an interesting qualitative piece that speaks to the universality of healthcare worker experience during the COVID-19 pandemic that goes beyond Italy's borders. Thank you for sharing. I'd like to see this point highlighted a bit more in the discussions/conclusions section. 

Round 2

Reviewer 1 Report

Dear authors thank you for replying to all the comments suggested above.

The paper in current form is improved, please find below minor revisions.

  • 2.1 Study design and sample: please add information about research design (for example, is it a qualitative research?).  
  • 2.1 Study design and sample, ethical committee: for this type of study and for nature of collected data, it would have been better to have the ethics committee approval. Therefore, for this reason, I suggest to only report the approval of the Director General (it would be better not to say that the approval of the ethics committee was not necessary, as this is not the case).
  • Results, title of Table 1: “Characteristics of the eligible sample (N=509; mean age=48,07 years; SD=10,97)” please remove information about age, the term eligible is confusing, consider a simpler title (example: Participant characteristics (n = 509) .

Reviewer 2 Report

I still cannot get a clear sense of the purpose of the study and how it was achieved by way of clear open ended questions. Perhaps, this would be clearer if reported alongside the quantitative results. There are many limitations to qualitative online methods which are not considered in the manuscript.
